# Optimization of In-Situ Exosome Enrichment Methodology On-a-Chip to Mimic Tumor Microenvironment Induces Cancer Stemness in Glioblastoma Tumor Model

**DOI:** 10.3390/cells14090676

**Published:** 2025-05-06

**Authors:** Saleheh Saffar, Ali Ghiaseddin, Shiva Irani, Amir Ali Hamidieh

**Affiliations:** 1Department of Biology, Science and Research Branch, Islamic Azad University, Tehran 1477893855, Iran; salehesaffar@gmail.com (S.S.); shi_irani@yahoo.com (S.I.); 2Department of Chemistry, Michigan State University, East Lansing, MI 48824-1322, USA; 3Institute for Stem Cell Research and Regenerative Medicine, Tehran University of Medical Sciences, Tehran 1419733151, Iran; 4Department of Anatomical Sciences, Faculty of Medical Sciences, Tarbiat Modares University, Tehran 1411713116, Iran; 5Pediatric Cell and Gene Therapy Research Center, Gene, Cell & Tissue Research Institute, Tehran University of Medical Sciences, Tehran 1419733151, Iran; aahamidieh@tums.ac.ir

**Keywords:** tumor microenvironment, exosome enrichment, response surface methodology, microbioreactor, glioblastoma, cancer stemness, optimization

## Abstract

Understanding cancer etiology requires replicating the tumor microenvironment (TME), which significantly differs from standard in vitro cultures due to nutrient limitations, acidic pH, and oxidative stress. To address this, a microfluidic bioreactor (µBR) with an expanded culture surface was designed to optimize exosome enrichment and glioblastoma cell behavior. Using response surface methodology (RSM), key parameters—including medium exchange volume and interval time—were optimized, leading to about a six-fold increase in exosome concentration without artificial inducers. Characterization techniques (SEM, AFM, DLS, RT-qPCR, and ELISA) confirmed significant alterations in exosome profiles, cancer stemness, and epithelial-mesenchymal transition (EMT)-related markers. Notably, EMT was induced in the µBR system, with a six-fold increase in HIF-1α protein despite normoxic conditions, suggesting activation of compensatory signaling pathways. Molecular analysis showed upregulation of *SOX2*, *OCT4*, and *Notch1*, with SOX2 protein reaching 28 ng/mL, while it was undetectable in traditional culture. Notch1 concentration tripled in the µBR system, correlating with enhanced stemness and phenotypic heterogeneity. Immunofluorescent microscopy confirmed nuclear SOX2 accumulation and co-expression of SOX2 and HIF-1α in dedifferentiated CSC-like cells, demonstrating tumor heterogeneity. These findings highlight the µBR’s ability to enhance stemness and mimic glioblastoma’s aggressive phenotype, establishing it as a valuable platform for tumor modeling and therapeutic development.

## 1. Introduction

The lack of a comprehensive in vitro tumor model is perhaps the deepest deficit in the realm of cancer study and therapeutic development. Animal models [1,2], even patient-derived xenograft (PDX) models [3,4], suffer from humane considerations and lack accuracy due to genotypical and, more importantly, immunological differences between species. Additionally, they offer very low throughput feasible assays.

Recent elegant studies on in vitro 3D tumoroid models derived from patient biopsies [5,6] in microfluidic systems [7,8] to simulate the tumor microenvironment (TME) [9,10] using geometric varieties in the bioreactor with different mass transfer strategies [11,12], such as air-liquid interface feeding systems [13,14], have shown promise. However, there is still no reliable in vitro tumor model available to researchers in this field. It is evident that a system intended to integrate all these advancements requires sophisticated instrumentation and specialties. Nevertheless, tumor microenvironment mimicry remains the deepest common root of all these efforts.

Access to an in vitro tumor model that encompasses all features of a tumor within the body is limited by several constraints. Among them, the chemically complex and heavily dense tumor structure is the most significant [15,16]. It is believed that cancer cells, through the TME—a chemically complex medium [17] inundated with chemokines [18] and extracellular microvesicles [19]—communicate with other cells both in their vicinity and distally to create an environment favorable for cancer progression [20].

The complexity of the TME is so extreme that it often sends mixed signals. For example, inducing angiogenesis [21], a cancer hallmark [22,23], results in incomplete tumor microvasculature formation, which is incapable of sufficient oxygenation of the tumor cells. Consequently, many tumor cells suffer from hypoxia [24] to such an extent that the hypoxic condition becomes a preconditioning for inducing tumor aggressiveness [25]. In many research projects, conditioned medium has been used to mimic the TME [26,27], and in others, TME simulation is targeted using microfluidic [28] or 3D tumoroids [29,30]. However, tumor models still cannot behave the same way they do in patients’ bodies.

We hypothesized that in either microfluidic systems or traditional tumor cell culture in conditioned medium, the TME that cancer cells try to provide for themselves has a far lower concentration of chemokines secreted by the cells compared to the native cancer cells within the TME in vivo.

To introduce a quantifiable parameter as a criterion for microenvironment enrichment in the μBR model, the exosome content of the culture medium was selected. This choice was based on a growing body of research highlighting the role of extracellular vesicles—particularly exosomes—as key mediators in cancer progression, due to their ability to carry a complex cargo of diverse chemicals, RNAs, and functional proteins that facilitate intertumoral communication [31,32,33,34].

To test this hypothesis, glioblastoma multiforme (GBM) cancer was modeled in a micro-bioreactor (μBR) in 2D culture without conditioned medium. The elimination of all systemic complexities, such as 3D organoid culture, intricate geometry, or engineered feeding systems like air-liquid interface, not only makes the tumor model more accessible in tumor research labs but also clarifies that controlled-TME condition is the essence of in vitro tumor modeling. To address all aforementioned factors in one simple model, GBM cells were cultured in a vast surface area in a microfluidic bioreactor, where the TME cultivation conditions were optimized via response surface methodology (RSM) and compared with traditional cell culture.

## 2. Materials and Methods

### 2.1. Study Design

#### 2.1.1. Estimating the Local Concentration of a Chemical Secreted from Tumor Cells Within the TME

Understanding that the thermodynamic activity of almost all cell products is strongly influenced by their effective concentrations in the TME [35,36], we performed basic measurements on the spatial conformation of cancer cells in the TME. The distance between cancer cells in a GBM solid tumor is less than 2 µm, with cell diameters ranging between 15 to 20 µm and a height of 10 µm. This means most of the volume of the solid tumor is occupied by cells. Thus, at best, about 29% of the volume of a solid tumor is interstitial space, which acts as a TME reservoir extracellular matrix (See Appendix A).

We measured the concentration of TGFβ in a tumoroid by ELISA (20 ng mL^−1^) without lysing the cells, assuming all the growth factor was in the TME. Based on this and knowing that only 29% of the total volume is allocated for TME, the concentration of TGFβ spiked to about 306,513 ng mL^−1^, which is 100,000 times higher than the concentration of measured TGFβ in traditional cell culture (see Appendix A for the calculations, (Appendix A)).

This can be one of the main reasons that in vitro models in traditional plate culture cannot mimic the heterogeneity of a solid tumor. The conditioned medium provides biochemicals and growth factors in the range measured via ELISA (in the TGFβ sample, about 20 ng mL^−1^), and the same concentration penetrates between the tumor cells, which is still far less than the secreted autocrine cytokines and chemokines trapped in the TME by several thousand folds. Although 3D cancer models show very promising results in mimicking solid tumor behavior, they still suffer from a lack of mass transfer limitations in the core of the tumoroids, leading to many abnormalities in the culture condition in the inner layers of the 3D models. For successful 3D culture and to overcome mass transfer limitations, it is necessary to immerse the 3D models in fresh, well-oxygenated medium, which not only differs from the medium quality in the center of the sphere but also continuously refreshes the medium, leading to diluted biochemical concentrations and lower thermodynamic activity.

Having observed these two extremes, GBM in vitro modeling falls between two boundaries. On one extreme, researchers can refresh the medium of the model regularly, keeping all the cancer cells thriving at the expense of losing autocrine effects and diluting out the TME component’s concentration in each medium refreshment cycle.

On the other extreme, lowering the medium refreshment and reducing the volumetric flow rate keeps the self-secreted biochemicals in close proximity to the cancer cells, mimicking the TME at the expense of accumulating waste materials, which are mostly toxic to the cells, and limiting nutrient and oxygen availability, not much similar to a native TME.

This setup was designed to cultivate a significant number of GBM cells in a 2D format, keeping them in close contact with nutrients and containing more than 4 × 10^5^ cells at about 75% confluency in a vast culture surface (area = 1.8 cm^2^) (Figure 1). The μBR chamber height was limited to 50 μm using soft lithography techniques for microfluidic device fabrication. We highly recommend reading the study design section represented in Appendix A.

Lowering the height in the μBR chamber limits the μBR volume to less than 9 µL. Using the device, we were able to dilute all biochemicals secreted from more than 4 × 10^5^ GBM cells in just 9 µL, meaning the auto-secreted biochemicals are diluted in about 100 times less culture medium compared to a standard 96-well plate culture. Lowering the resident volume of the medium by about 100 times is a significant improvement to mimic the TME, but still, the concentration of self-secreted chemokines from the cells in vivo is much higher.

The other parameter that should be managed is the feed flow rate. To allow the system to accumulate the cell’s secreted biomolecules and components in the μBR chamber, a semi-fed-batch strategy has been chosen for medium renewal. In the semi-fed-batch strategy, the medium of the μBR is partially replaced with fresh medium at certain time points. Two reservoirs were considered for medium directly connected to the μBR chamber. Basically, the feed flow rate is comprised of two components: feed volume and the time that the corresponding volume is fed ([volume]·[duration time]^−1^). In the feeding strategy and related mathematical modeling, the flow rate is decomposed into components through which researchers adjust nutrient, oxygen, waste accumulation, and pH of the TME, aiming for the lowest possible medium exchange to achieve the highest accumulation of cell-secreted bioactive materials yet in close similarity to pathophysiological conditions.

To this end, Response Surface Methodology (RSM) was employed, covering essential aspects needed to govern the μBR. Since all the effector variables such as glucose concentration, soluble oxygen, pH, and microelements are provided in the feed, the μBR operation and the mathematical model of the μBR were simplified into two variables: the volume of refreshing medium feeding to the system via the semi-fed-batch strategy and the interval time between each medium exchange. To close the feedback control system to run the RSM model, three responses were selected to represent the real-time condition of the microbioreactor: glucose concentration, cell proliferation status, and exosome content of the μBR.

#### 2.1.2. Glucose Concentration and Cell Proliferation

The glucose concentration delegates nutrients and all other components that have to be provided through feeding. Cell proliferation status represents the pathophysiological condition of the cells in the tumor model. By monitoring cell proliferation, the model can determine how close we are to the lower limit boundary condition. As mentioned earlier, the μBR operates between two boundaries, with the lower one being the reduction of medium exchange and volume to a point where the cells cannot survive due to lack of nutrients and toxic material accumulation. Since cancer cells are proliferative in a normal pathophysiological condition, if the proliferation status evaluated by live/dead cell assay is not declining, the model recommends reducing the volume of medium exchange and deferring the time of the exchange.

#### 2.1.3. Exosomes

Exosomes were believed, at moment of discovery, to cargo the waste out of the cytoplasm [37,38]. Since then, day by day the crucial role of exosomes becomes clearer in terms of cell signaling pathways and cell-cell communication [39]. Each exosome carries several miRs, siRNAs, functional proteins and peptides enveloped in a bilayer membrane and secreted out of cell via exocytosis [40]. Therefore, exosomes are the best candidates for representing the autocrine effect in TME, because the exosomes not only represent a wide range of cell secreted biomolecules in a package, but they also represent waste accumulation in the TME which is the negative side of declining the culture medium renewal. Through measuring the exosome content gradient in the TME of the tumor model, the model can confirm the accumulation of the cells secreted biochemicals. As it is demonstrated in Figure 1f, hypothetically in each cycle of medium refreshment, most of the exosome content is removed while in the semi-fed-batch strategy just the amount that retrieves physiological condition is exchanged.

### 2.2. Materials

A PDMS elastomer kit was purchased from Dow Corning Inc. (Midland, MI, USA). Photoresist SU8-2050 and its developer were obtained from MicroChem Corp. (Westborough, MA, USA). Low-glucose DMEM, penicillin-streptomycin, and fetal bovine serum (FBS) were purchased from Gibco, Thermo Fisher Scientific (Waltham, MA, USA). Acridine orange (AO), phosphate-buffered saline (PBS) solution, and paraformaldehyde were obtained from Merck KGaA (Darmstadt, Germany). The MTT assay kit and 3-aminopropyltriethoxysilane (APTES) were purchased from Sigma Aldrich, Merck KGaA (St. Louis, MO, USA). DAPI (4′,6-diamidino-2-phenylindole, dihydrochloride) and the antibodies Notch1 (Cat# EHNOTCH1) and β-Catenin (Cat# KH01211) were obtained from Thermo Fisher Scientific (Waltham, MA, USA). The HIF1 antibody (Cat# H-3091K) was purchased from Abnova (Taipei, Taiwan), and ZEB1 (Cat# GWB-KXYPE) from Aviva Systems Biology (San Diego, CA, USA). The SNAIL antibody (Cat# abx153126) was obtained from Abbexa Ltd. (Cambridge, UK). The following products were obtained from Abcam (Cambridge, UK): antibodies SOX2 (Cat# ab245708), E-Cadherin (Cat# ab233611), N-Cadherin (Cat# ab244512), Vimentin (Cat# ab246526), and the glucose assay kit (Cat# ab65333). Human multiform glioblastoma cells (U87 MG) were purchased from the Iranian National Cell Bank, Pasteur Institute (Tehran, Iran).

### 2.3. Microfluidic Device Fabrication

The microfluidic device was designed in Corel (version 22) and fabricated by a standard soft lithography technique. Cast molds were prepared by photoresist SU8-2050 on glass by conventional photolithography. The mixture of poly (dimethyl siloxane) (PDMS) pre-polymer solution and curing agent at ratio of 10:1 (*v v*^−1^) was poured onto the patterned glass to mold the microfluidic devices [39]. The chamber diameter was considered 15 mm and the fluid channel height was 50 μm. The inlet and outlet were created with a 3 mm biopsy punch. The cured PDMS structure was attached to two pieces of polystyrene sheets using screws (Figure 1a–e). To sterilize the device components, they were incubated in 70% ethanol for 10 min, rinsed with pure ethanol, and dried, and finally exposed to ultra-violet (UV) light for 30 min.

### 2.4. Experimental Design

Thirteen sets of experiments were designed and analyzed using a central composite design (CCD-response surface; Design Expert Software (DOE; version 11.0.3.0, Stat-Ease, Minneapolis, MN, USA) as shown in Table 1. Two variables, the time interval for the replacement of fresh culture medium (A: Time) and the replaced volume of culture medium (B: Volume), were selected to optimize the concentration of enrichment exosome. The volume range selected before optimization was from 150 to 400 μL and for the time between 5 and 15 h. The responses were collected in terms of these variables including cell proliferation, exosome content, and glucose content of the cell culture medium. After performing 13 stages of testing, by specifying the range of variables and responses through numerical optimization, the optimal values of each response were determined in the best case and compared with the control group, which was traditional cell culture. Statistical analysis using ANOVA with DOE software was used to estimate the optimal culture conditions for the maximum concentration of exosomes enriched by cells. The coefficients in the first and second-order polynomials were calculated using multiple regression analysis of the obtained results experimentally (See Equation (1)).
(1)$$Y=b_{0}^{\prime }+\sum_{i=1}^{n}b_{i}X_{i}+\sum_{i=1}^{n}b_{ii}X_{i}^{2}+\sum_{i=1}^{n}.\sum_{j\geq 1}^{n}b_{ij}X_{i}X_{j}$$
where *Y* is the predicted response, $b_{0}^{\prime }$ is the constant coefficient, *b_i_* is the linear coefficient, *b_ij_* is the interaction coefficient, *b_ii_* is the quadratic coefficient, and *X_i_* and *X_j_* are coded values [41].

### 2.5. Cell Culture and Sample Preparation

Human multiform glioblastoma cells (U87 MG) were cultured in T25 flasks in a medium containing low-glucose DMEM supplemented with 15% (*v v*^−1^) FBS, and 1% penicillin-streptomycin (104 U mL^−1^). All cells were treated under sterile tissue culture hoods and maintained in 5% CO_2_ humidified incubator at 37 °C. The culture medium was replaced every 24–48 h until cells reached 70–80% confluence. To culture the cells inside the bioreactor chamber, we assembled the device package with the cell loading inlet and outlet open, as shown in Figure 1e.

Before loading the cells onto the chip, to sterilize the device components, two pieces of polystyrene were incubated in 70% ethanol for 10 min, rinsed with pure ethanol, dried, and finally exposed to ultra-violet (UV) light for 30 min. After the ethanol solution was rinsed out with phosphate-buffered saline (PBS) solution, the culture chambers were treated with culture medium solution for 24 h to enhance cell adhesion to the culture chamber. The cells were suspended in the flask, and after centrifugation, forming a pellet, and dissolving it in a certain amount of culture medium, they were gradually transferred from the inlet to the culture chamber. The device was pumpless, and adjusted to show a free flow of medium from inlet into the culture chamber and from chamber to outlet. Before optimization, thirteen cell culture phases were carried out inside the microfluidic device following Table 1, and the exhausted medium partially was removed from the device in accordance with the schedule created in the RSM approach. This allowed for the optimal response points to be obtained.

### 2.6. Glucose Assessment

The glucose levels in two groups, optimized micro-bioreactor (optimized µBR) and the traditional cell culture, were detected using glucose assay kit (Cat# ab65333; Abcam, Cambridge, UK). We adjusted the volume of culture media to 50 µL per well with Assay Buffer II to Glucose Assay Buffer. All the steps were conducted as the manufacture protocol described.

### 2.7. Tortuosity Index

The tortuosity index was calculated using Image J software (version 1.53c; National Institutes of Health, Bethesda, MD, USA) separately for two groups: the optimized micro-bioreactor group and the control group. For each group, on the first day when the cell morphology was completely round, we considered a circle tangent to the circumference of the cell’s body. This process was repeated for a certain number of cells, and the average area of these circles was calculated. On the seventh day, when the cell morphology showed spread out appendages, we repeated the process of drawing circles that completely covered the cell for the same number of cells. We averaged the area of these circles. Finally, the tortuosity index for each of the two groups (microfluidic device and traditional cell culture) was obtained by dividing the average area of circles on the seventh day by the circle’s area on the first day [42].

### 2.8. Viability Assay

Live-dead assay: The viability of the cells was evaluated through acridine orange (AO) staining and MTT (Thiazolyl Blue Tetrazolium Bromide) assay in both groups. The cells were fixed with 4% paraformaldehyde and then stained with acridine orange (AO) to evaluate the number of apoptotic cells in each group.

An aliquot of acridine orange (AO) reagent (1 µL of 2–5 µg mL^−1^) was added to each group and incubated for 10 min at room temperature [43]. AO is a membrane-permeable nuclear dye that causes cells with RNA/ssDNA to have a red color and dsDNA to have a green fluorescence. The samples were washed with PBS, and images were captured using a fluorescence microscope (Eclipse Ti2-E; Nikon Corporation, Tokyo, Japan).

MTT assay: To perform MTT assay, after culturing of the cells in the optimized µBR and traditional cell culture (T25 flask) for 0, 1, 3, and 7 days, MTT solution was added and incubated for 3 h. After discarding the supernatants, the sediments dissolved in dimethyl sulfoxide. Samples were shaken for 15 min in the dark. The absorbance of the extracted solution was measured at 570 nm using an ELISA reader instrument (Epoch, BioTek Instruments Inc., Winooski, VT, USA) [44]. Finally, µBR and traditional cell culture data were compared.

### 2.9. Scanning Electron Microscopy (SEM)

The exosome samples were deposited on the activated coverslip surface with 3-aminopropyltriethoxysilane (APTES) and dried overnight and then washed using PBS and then with distilled water. The samples were coated with a gold layer using a sputter coater, and images were captured using a scanning electron microscope (AIS2100; Seron Technologies Inc., Uiwang-si, Gyeonggi-do, Republic of Korea) [45].

### 2.10. Atomic Force Microscopy (AFM)

After drying the exosome samples on APTES-activated cover slips, they were left to air dry at room temperature and washed with PBS followed by distilled water [46,47]. Images were acquired using an atomic force microscope (Solver PRO; NT-MDT Co., Moscow, Russia). The images were analyzed using NOVA software (version 1.26.0.1443, NT-MDT Co., Moscow, Russia).

### 2.11. Dynamic Light Scattering (DLS)

A dynamic light scattering test was applied to determine the particle size distribution. Enriched exosome cell culture samples were diluted with PBS to 2% to prevent multiple scattering. The solution was poured into specialized cuvettes and analyzed using a dynamic light scattering and zeta potential analyzer (SZ-100Z; Horiba Ltd., Kyoto, Japan) [48].

### 2.12. Immunocytochemistry (ICC) and Analyze Marker Expression

The samples from all study groups were fixed on the µBR surface in 4% paraformaldehyde for a few minutes at room temperature, then the PDMS chamber of the µBR was removed. All samples were washed with PBS three times for 1 min and incubated with the primary antibodies SOX2, Vimentin, E-Cadherin (all from Abcam), and HIF-1α (from Abnova) for 1–2 h at room temperature or overnight at 4 °C. Samples were then washed with wash buffer and PBS three times for 5 min, followed by blocking in 0.5–1% BSA for 30 min at room temperature. After preparation of appropriate dilutions of fluorochrome-conjugated secondary antibodies in 1% BSA, the samples were incubated with the secondary antibody dilution for 2 h at room temperature in the dark, then gently washed in PBS three times for 5 min each. Finally, immunofluorescence images were captured using a fluorescent microscope (Eclipse Ti2-E, Nikon) on days 1 and 7.

### 2.13. Real-Time RT-PCR

To evaluate gene expression, total RNA was extracted in the µBR and traditional cell culture samples on days 0 and 7. Total RNA concentrations were determined using a NanoDrop spectrophotometer and absorbance was measured at 260 nm (Thermo Fisher Scientific, Waltham, MA, USA). Complementary DNA (cDNA) was synthesized according to the manufacturer’s instructions. Then RT-qPCR was performed to analyze the expression of *BAX* (as an apoptotic marker), *BCL2* (as an anti-apoptotic marker), *Ki67* (as a proliferation marker), *OCT-4* and *SOX2* (as stemness gene), *NF-κB* (as an inflammation factor and coactivator of CSC and EMT related genes), *E-Cadherin*, *N-Cadherin*, *Vimentin*, *SNAIL* and *ZEB1* (as EMT epithelial related markers), *β-Catenin*, *STAT3*, *Notch1*, *TGFβ*, *EGF* and *SMAD* (as crucial regulator genes in stemness and EMT related genes) and *β*-*actin* (as a housekeeping gene). Primers for RT-qPCR were designed using Primer3 (v2.3.6), and their specificity was validated against NCBI BLAST databases (nr/nt and wgs) [https://blast.ncbi.nlm.nih.gov/Blast.cgi, accessed on 25 April 2025]. The sequences of the primers are listed in Table 2. All experiments were conducted in triplicate. qRT-PCR results were analyzed using REST 2009 software (Qiagen, Hilden, Germany), and fold changes in gene expression were calculated using the 2^−ΔΔCT^ method [49].

### 2.14. Enzyme-Linked Immunosorbent Assay (ELISA)

The cultured medium of U87MG cells was collected. The cultured medium of U87MG cells was collected. The expression levels of the following key proteins, Notch1 (main player of important pathway in triggering EMT), HIF1 (Hypoxia-Inducible Factor 1 is known for its involvement in promoting aggressiveness in cancer cells, regulates cellular response to hypoxia and can enhance invasion potential by inducing various target genes related to angiogenesis, epithelial-mesenchymal transition (EMT), and extracellular matrix remodeling), ZEB1 (as EMT marker, related to dedifferentiation of the glioblastoma cells to mesenchyme which convert cells to a motile type and its expression is associated with higher grades of malignancy, tumor progression and invasion), SOX2 (as stemness marker has been linked to enhanced invasiveness in various cancer types, potentially through EMT induction and interaction with other signaling pathways), β-catenin (a component of the cell-cell adhesion complex and of the canonical Wnt pathway, regulates proliferation, adhesion, and migration in different cell types, as crucial regulator in stemness and EMT related genes), E-cadherin (an epithelial marker), N-cadherin (cancer epithelial marker), Vimentin (mesenchymal marker), SNAIL (as EMT marker constitutes a master switch that directly represses the epithelial phenotype) in Glioblastoma cell line (U87) culture under optimized condition in µBR vs. Traditional cell culture supernatant were measured using ELISA kits on days 0 and 7, following the manufacturers’ protocols: Notch1 and β-Catenin (Thermo Fisher Scientific), HIF1 (Abnova), ZEB1 (Aviva Systems Biology), SOX2, E-Cadherin, N-Cadherin, Vimentin (all from Abcam), and SNAIL (Abbexa).

### 2.15. Statistical Analysis

Results are expressed as mean ± standard error (SE) and analyzed by the GraphPad Prism 9.4.0.673 (GraphPad Software Inc., La Jolla, CA, USA). Technical as well as biological triplicates of each experiment were performed. Comparison between two groups was performed by Student’s *t* test. Multiple group comparisons were determined using one-way analysis of variance. Values of *p* < 0.05 were considered statistically significant (* *p*-values < 0.05, ** *p*-values < 0.01, *** *p*-values < 0.001 and **** *p*-values < 0.0001).

## 3. Results

Using response surface methodology (RSM) to develop a numerical model of the tumor microenvironment (TME), we found that reducing the volume of the cell culture environment, although it mimics the TME, leads to cell starvation and the accumulation of cell waste, causing cell toxicity and death. Therefore, it is standard practice to keep the cells in fresh medium in vitro. To find the optimal point for medium exchange rate (volume and interval time of medium renewal), RSM was applied to three responses—cell proliferation, glucose content, and existing exosome number in the medium—using Design Expert software to model the microbioreactor environment behavior. The conditions were monitored to maintain the glucose content in a predefined range, not lower than 40 mg/dL, and to keep the cells proliferative and far from starvation. On the other hand, exosome content was used to represent the TME’s components, which we aimed to maximize to approach the natural TME.

Technical details about RSM modeling and optimization are provided in the extended data section, and the experimental points of the study used for the modeling are summarized in Table 1 in the method section. Based on mathematical modeling of the responses as a function of the medium exchange rate, an optimized point was defined. Figure 2a shows that exosome content ranges between 27 and 178 × 10^3^ exosomes per µL while the glucose is within the standard range and cells are proliferative. It also demonstrated that all high exosome contents occur close to the border limits of other responses and parameters, confirming the hypothesis of the study. In Figure 2b–e, the sensitivity of each response against the other responses and parameters is shown. All three response surfaces are drawn as a function of medium volume and interval time of exchange in 3D graphs. As shown in Figure 2c, the model predicts that even 189 × 10^3^ exosomes/µL is accessible within the defined condition, which was later confirmed experimentally in the optimal condition experiment (Figure 3b) with 12-h intervals and 150 µL volume of the renewed medium.

### 3.1. Culture Condition Characterization

The culture condition is characterized by measuring the glucose content, which fell between 30 to 90 mg/dL, and the proliferation status of the culture using a live/dead cell assay (Figure 3g,h). Acridine Orange (AO) has the unique property of emitting green when it binds to double-stranded DNA and turning orange to red when disintegrated genomic strands unwind to single-stranded DNA, demonstrating the life cycle standpoint of the cells [50]. By reducing the volume of the medium exchange from 400 µL to 150 µL and increasing the time intervals from 5 h to 15 h, the cells were kept in a condition where more than 95% (Image J analysis) of them emitted bright green in AO fluorescent microscopy while they were still proliferative. Reducing the volumetric rate of the feed resulted in lowering the proliferation rate by about 29%, showing the effectiveness of the feeding strategy. At the same culture and time point, the glucose content also dropped to about 40 mg/dL. Moreover, at the 15-h time point of medium exchange, the phenol red pH indicator of the medium turned orange, indicating that the pH was also declining by that time. Governing the µBR in this condition for 7 days of culture led to significant changes in the tumor model on many levels, including transcriptomic, proteomic, and even phenotypic and morphological changes of the cells without adding any supplements to the medium or using conditioned medium, which will be discussed later.

### 3.2. Exosome Characterization

Exosome content was selected as the TME enrichment criterion. Isolated exosomes were characterized using electron microscopy (Figure 3c) and atomic force microscopy (Figure 3d–f). Most of the isolated nanoparticles extracted from discharged µBR medium ranged between 100 nm and 300 nm, as confirmed by dynamic light scattering measurement (Figure 3i). The growth rate of the exosomes, in terms of number in a controlled volume, was measured on day one and day seven of cancer cell cultivation in the µBR with different medium renewal intervals (Figure 3j). The results clearly indicate a significant increase in exosome content by lowering the number of medium exchanges. By increasing the interval time of medium exchanges from 5 h to 6.5, 10, 13.5, and 15 h, as suggested via RSM, an almost linear increase was observed in exosome number, starting from 50 × 10^3^ exosomes/µL (at 5-h intervals) to about 178 × 10^3^ exosomes/µL at 13.5-h intervals (Figure 3j). The accumulation of exosomes in the µBR chamber during the culture was also confirmed by comparing the exosome numbers between day 1 and day 7 (Figure 3j), which confirms the effectiveness of the feeding strategy and the model.

During all trials for increasing the number of exosomes, the parallel culture in µBRs was monitored for cell viability and cytotoxicity effects of partially remained exhausted medium in the µBR chamber, which always kept the cells fresh (always more than 95% viable cells) (Figure 3g,h).

### 3.3. Cancer Model Optimization of TME Using RSM

After extracting data from different time intervals and various volumes of medium exchange, the data were fed into RSM, and the model was optimized for the highest possible exosome content while keeping the other two readouts within the predefined ranges. As shown in Figure 3a,b, the highest desirability of the model condition occurs at the lowest borderline medium exchange volume of 150 µL. With a lower amount of medium exchange, the pH of the remaining medium cannot be returned to the physiological range of the cell culture. Through optimization, we found the borderline cultivation condition in which we enrich the medium with cell-secreted materials at the highest possible concentration while keeping the cells in normal cell culture conditions.

The optimized model predicts (Figure 3b) that using a 150 µL volume of medium exchange at 15-h intervals will result in a sharp fall in cell viability, which is consistent with our observations. Any time close to or more than 15-h intervals with low volume exchange results in acidic pH and probably the release of lytic enzymes in the TME, which results not only in cell cytotoxicity but also in digesting the exosomes detectable in dynamic light scattering measurements.

### 3.4. Comparing the Behavior of Cells in Optimized Culture Conditions in μBR with Traditional Cell Culture

After running the optimization process on the model, which defined the optimized condition at 150 µL for the volume of medium renewal and 12.0 h for medium exchange intervals and augmentation runs, the tumor model was run in the μBR and in a petri dish standard cell culture to demonstrate whether the controlled TME can induce any pathway in glioblastoma cells. The experiments revealed that in the μBR glioblastoma cell culture, the cell growth rate was hampered by more than 33% in 7 days (Figure 4a), indicating suppression in the cell line artificially overexpressed proliferation factors. The results were later confirmed using image analysis. The decrease in proliferation rate in the optimized μBR was 10-fold higher compared to traditional cell culture (Figure 4b,d). The glucose level of the discharged medium from the μBR was consistently about 40 mg/dL, while in traditional cell culture, the glucose content was about 100 mg/dL (Figure 4c). The glucose level represents the inlet feed components. By maintaining it at the same level in all measurement points for 7 days, the feed condition of the μBR chamber can be assumed to be in a steady-state condition.

There were no significant differences between the shape and size of the exosomes extracted from the two cell cultures on different days (Figure 4e). Although the glucose content of the medium was 2.5-fold lower than the standard condition and a steep decrease was observed in cell proliferation, there were no signs of apoptosis, such as DNA fragmentation, apoptotic body formation, blebbing, or even orange light from acridine orange staining (Figure 4d). Interestingly, the accumulated exosome content after 7 days in the optimized μBR was more than 400% higher than in traditional cell culture, showing less than a 10% difference from the predicted number from the RSM model. Given the success of the design in terms of accumulating self-secreted materials from the cells in the TME, the samples were collected for study at the transcriptomics and proteomics levels.

### 3.5. Glioblastoma Model On-a-Chip

The U87 glioblastoma cell line, a neuroepithelial cancer cell line well-characterized for its genotypic and phenotypic stability, was seeded in the µBR under optimized conditions. Several scientific reports have attempted to induce different fates in cancer cells, both cell lines and primaries, using either conditioning medium [51] or gene manipulation [52]; in all of them, the cultivation with no treatment was observed as a control group in which the epithelial and proliferation markers are conserved. In this experiment, we optimized the condition as discussed previously, with no additional chemicals as inducers or inhibitors, and the incubator condition was standard (not hypoxic). The only effective variable was the mimicked TME.

Prior to monitoring famous pathways in the model, apoptotic and anti-apoptotic markers, *BAX* and *BCL2*, respectively, were evaluated using RT-PCR. Since glioblastoma cells are blind to the p53 apoptotic pathway [53], normal programmed cell death should not be detected during the culture. However, there was still a possibility that starvation from the optimized culture condition—running the µBR in lower boundary conditions in terms of nutrients—would cause overexpression of BAX, leading to inhibition of the BCL2 protein family and upregulation of apoptosis. Although both *BAX* and *BCL2* expression levels slightly increased in the µBR compared to traditional cell culture, the anti-apoptotic marker (*BCL2*) expression was about 40% higher than the apoptotic marker (*BAX*) in the µBR (see Figure 5). Therefore, the decrease in cell proliferation observed is not due to the activation of the apoptotic pathway, which is also consistent with cell morphology and AO staining. Moreover, the possibility of anoikic (detachment-inducible apoptosis) cell death was eliminated by performing an MTT assay (Figure 4a).

Knowing the neo-plasticity of the cells and that the apoptotic pathway is not activated, the dedifferentiation pathway was studied. U87 cells are neuroepithelial astrocytoma cells that are strongly positive for normal and cancer epithelial markers, *E- & N-cadherin*. Surprisingly, after one week of culture in the mimicry condition in the µBR, the expression of both genes was dramatically downregulated (Figure 5).

Losing epithelial fate, especially adhesion surface protein expression (E-cadherin and to some extent N-cadherin), was not only a sign of reviving the heterogeneity and invasiveness of glioblastoma cell lines but also could explain the reduction in cell proliferation in the µBR (Figure 4b). In fact, cell proliferation in the mimicked condition in the µBR was not decreased; rather, the cells lost their anchorage ability to the surface due to the repression of cadherin proteins; hence, they were washed off during medium exchanges. This observation can be a clue for further studies to mimic cancer invasion phenomena. To confirm the ability of cell proliferation in the TME-mimicked µBR, *Ki67*, a cancer proliferation gene, was monitored, which was consistent with observations, though it was about 11% lower than it should be compared to traditional cell culture conditions (Figure 5).

### 3.6. Epithelial-Mesenchymal-Transition Pathway Activation

To prove that the model is capable of inducing neo-plasticity through dedifferentiation of the glioblastoma cells by triggering the EMT pathway, the Vimentin level was first measured by ELISA (Figure 6). The Vimentin content in the lysate on day 7 was about 6-fold higher than it was on day one. Additionally, it was about 4-fold higher compared to the same condition in traditional cell culture. Conversely, the mRNA level of *Vimentin* transcribed in traditional cell culture was about 4-fold higher than in the µBR. To confirm the results from ELISA, an immunocytochemistry (ICC) assay was performed, confirming the presence of Vimentin in µBR-cultivated cells (see Figure 7 confocal microscopy images and related graphs). However, we still cannot explain what inhibited Vimentin protein synthesis in traditional cell culture despite the high level of *Vimentin* mRNA content, which needs further investigation.

### 3.7. EMT Pathway Is Triggered via a Variety of Axes

As depicted in Figure 5 and Figure 6, the EMT pathway is progressing through multiple axes. Accumulated TGF-β in the tumor microenvironment (TME) stimulates the *SMAD-Snail-ZEB1* axis, while the *Wnt-*β-catenin-*ZEB1* cascade is also ongoing. Additionally, epidermal growth factor (EGF) autosecretion can trigger a negative feedback control loop to activate the RAS/MEK signaling, leading to ZEB1 overexpression and inhibition of epithelial markers (Cadherin) [54,55]. The lowest rise in EMT transcriptomic factors (about a 150% increase) belongs to *NFĸB*, which can be interpreted as the absence of immune cells in the environment, as *NFĸB* autocrine effects are mostly heightened via immune cell inflammatory chemokines [56].

### 3.8. Enriched TME Stimulates HIF-1α Compensatory Pathway

When the model showed that the enriched TME can run the EMT pathway to dedifferentiate U87 cells into mesenchymal ones, one of the most important questions to be answered was whether EMT is induced by a compensatory signaling pathway. It is well established that hypoxia drives cells to EMT pathways through compensatory cascades initiated by expressing hypoxic inducible factor 1 protein (HIF-1α) [57,58]. In this study, we did not impose any hypoxic condition on the µBR chamber; however, surprisingly, HIF-1α protein concentration increased in the µBR group more than 6-fold (Figure 6). Dissolved oxygen concentration measurement in the µBR chamber was not accessible due to the limited volume of the chamber, but there was no sign of hypoxia or a toxic environment in the cell’s morphology. Glioblastoma with high levels of mesenchymal transcription factors is associated with a high proliferation rate and poor vascularization in the core of the tumor, which has always been the justification for the hypoxic microenvironment of the tumor [59]. Here, we observed that the U87 cells in the controlled microenvironment, even in normoxic conditions, can accumulate HIF-1α proteins up to a threshold that promotes EMT.

### 3.9. Optimum Controlled Microenvironment Revives Neoplasm in the Cell Line

There are studies that have developed methods for dedifferentiation of glioblastoma cells into cancer stem cells (CSCs). Afify et al. published a method for producing CSCs using extracted glioblastoma exosomes [60]. Sun et al. have shown that Notch protein carried in glioblastoma-extracted exosomes plays a crucial role in driving mature cancer cells to CSC-like cells [61]. The presence of the stemness markers (*SOX2* and *OCT4*) and Notch1 protein were tested in the transcriptome and proteome using RT-PCR, ELISA, and ICC. Both *SOX2* and *OCT4* gene expression increased about 2-fold (Figure 5). SOX2 protein accumulation in the mimicked TME was about 28 ng/mL, where it was barely detectable in traditional cell culture medium. Notch1 protein concentration in spent medium after day 7 of culture in optimized µBR culture was about 1250 pg/mL, whereas its initial concentration in the µBR was about 400 pg/mL. During the same time of cultivation of the same cells in traditional cell culture, there was no significant change in this protein concentration (Figure 6—Notch1).

Immunofluorescent microscopy confirmed SOX2 protein accumulation in the nuclei of the cells in the µBR group, while it was negligible in the other group (Figure 7a–c). Moreover, Figure 7g–n demonstrates the HIF-1α marker in (image l, n) the cells, which is not detectable in the other group nor on the first day of either one. Figure 7j,m shows the presence of SOX2 and HIF-1α in dedifferentiated CSC-like cells (the bright yellow nuclei—SOX2—with red/orange cytoplasm for HIF-1α) and the cancer cells (the green cells in the background) in one frame, which is a sign of reviving the neoplasm or heterogeneity of the TME (see Appendix A for the additional photos, Appendix A).

### 3.10. Mimicked TME in µBR Changes the Cancer Cell’s Phenotype

One of the most interesting observations during the use of this experimental setup is the visible transformation in the morphology of the cells starting at day 3 of culture. As represented in Figure 8a–j, the glioblastoma cells, after seeding and attachment to the cultivation surface, kept their round shape with very limited polarization in the early days of µBR culture and traditional cell culture conditions. However, from day three until day seven in optimized cell culture in the µBR, the cells became highly polarized and elongated (see Figure 8c,h). The cultured cell’s phenotype in the µBR was similar to neuronal rosettes (Figure 8g), even though the culture was on a 2D surface (see Appendix A for the additional photos, (Appendix A)).

## 4. Discussion

Observing a low expression of the *E-cadherin* gene was a promising indication that the TME could induce the epithelial to mesenchymal transition (EMT) pathway. ZEB1 (zinc finger E-box binding homeobox-1) is a DNA-binding protein that binds to the *CDH1* gene’s promoter, suppressing cadherin surface proteins [62]. Overexpression or accumulation of *ZEB1* mRNA (Figure 5) in the optimized culture in µBR, which was later reinforced by controlling the protein expression level using ZEB1 protein ELISA (Figure 6), explains the low expression of cadherin proteins and led us to further investigate EMT.

To elaborate on the capability of the tumor model in investigating the potential governing pathways, different cascades of the EMT pathway were evaluated. EMT can proceed through the *Wnt*/β-catenin/*ZEB1* [63] cascade or by activating *NF-kB*, mostly through inflammatory transcriptomic factors [64], or through the TGF-β/*Snail* axis [65]. EMT can also be promoted through immunogenic factors and chemokines like TNFα [66] and interleukins, which were not investigated here due to the lack of immune cells in the µBR. The schematic in Figure 8 demonstrates two different fates in green and magenta colors, which can be induced or inhibited by cues in the TME.

Afify et al. and Sun et al. have shown that isolated exosomes from spent cancer cell medium can induce cancer markers and cell transformation [60,61]. They also comprehensively investigated the Notch1 signaling pathway in turning on and off CSC-like differentiation or dedifferentiation through triggering the pathway by Notch1 present in cancer cell’s enriched exosomes. Yoshimoto et al. have shown that this phenotype transformation occurs during TGF-β dependent epithelial to mesenchymal transition [51], which is consistent with our results, where the tortuosity index (defined in the methods section) heightened by about 240%. They can induce the transition into cancer cells using conditioned medium with TGF-β supplement. Our results depict about 5-fold higher *TGF-β* gene expression in optimized µBR compared to traditional cell culture (Figure 5).

The phenotype change in the glioblastoma cells proves that naturally enriched TME in the µBR can denote heterogeneity of the tumor at the transcriptomics and proteomics levels. The expressed proteins and other autosecreted components can functionally promote changes both in desirable and malfunctioning ways, leading to a complex environment where cells sometimes detach, show a tendency to migrate, behave like proliferative ones, or even express markers irrelevant to their origin, which is called a heterogeneous tumor microenvironment depicted in this research.

## 5. Conclusions

Microfluidic device application in in vitro diseased models has been drawing researchers’ attention in the past two decades to address their concerns about mimicking pathophysiological conditions. Specifically, in the realm of in vitro tumor models, TME is the focal point of reasoning to apply microfluidic systems. This research has shown that although downsizing of the culture environment by microsystems is a leap towards TME simulation, the effective chemical activity of components in the interstitial space of tumor cells is significantly higher in native TME. Applying RSM in this research, we approached the highest possible accumulation of components in the TME, which is still lower than native TME in patients’ bodies.

The results enlighten that through a fed-batch strategy of medium renewal in µBR with a vast surface, the microsystem can revive the plasticity of the glioblastoma cell line with no additional conditioning components. To shed light on how far this plasticity can go, future investigations are needed, especially in some cases where long appendages in cells (axon-like) have been observed, suspicious of differentiation of CSC-like cells towards neurons. Another future direction can include single-cell RNA sequencing to elaborate dominant pathways beside native tumor biopsy culture, which can detect trace effective components that normally diluted out in the culture medium. The coculture of this model with immune cells, either brain microglial cells or chemokines secreted from blood-circulating immune cells, can also be highly interesting for a wide variety of research investigators.

## Figures and Tables

**Figure 1 cells-14-00676-f001:**
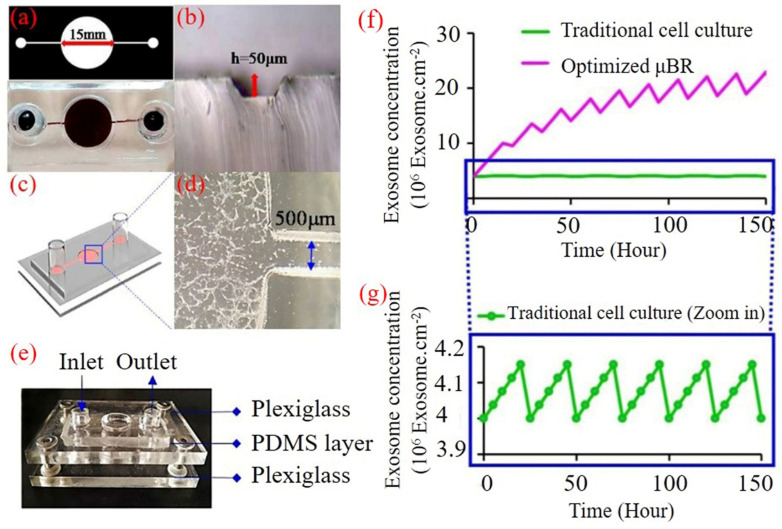
A microfluidic bioreactor with a vast surface for cell culture mimics tumor microenvironment. (**a**) Lithographic image of a two-channel microfluidic device for enrichment of exosome concentration, the high diameter (15,000 µm) of the middle chamber intended for cell culture. The limited height of the chamber (50 µm) in alliance with vast cell culture surface provides a culture condition with low volume of medium per cells which ends with higher concentration of cell secreted components including exosomes. (**b**) Cross section of PDMS micro-channel with a height of 50 µm. (**c**) Schematic design of various components of the microfluidic device. (**d**) Top view of the inlet channel with a diameter of 500 µm and part of micro-chamber where U87MG cells are cultured. (**e**) Microfluidic device with one inlet designed for cell and media flow and one outlet for discharge and collection of culture medium containing concentrated exosome. (**f**) Theoretical comparison of exosome concentration changes in traditional cell culture feeding and fed-batch medium exchange strategy in microfluidic device. (**g**) Enlarged image showing of exosome concentration changes in traditional cell culture with more details.

**Figure 2 cells-14-00676-f002:**
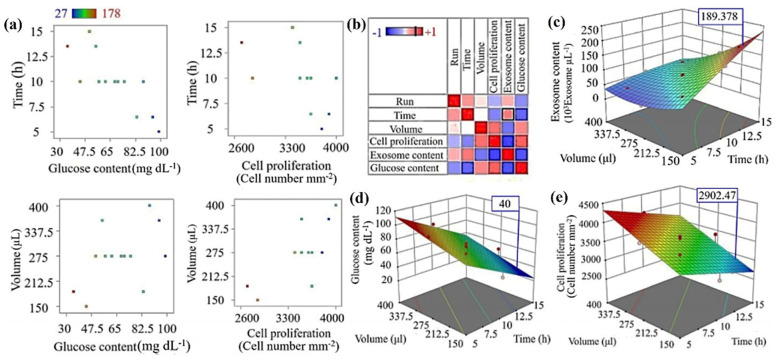
Response Surface Methodology (RSM) for mathematical modeling and optimization of Glioblastoma cell culture in the micro-bioreactor. Two variables, volume of exchange medium and interval time of medium renewal, which govern the feed rate (volume/time) have been hired as variables in RSM to optimize three responses, exosome content, glucose content, and cell proliferation. (**a**) The correlation between variables changes and responses in study points. (**b**) Exosome content sensitivity analysis correlated to the other variables and responses. (**c**–**e**) Exosome content, Glucose content, and cell proliferation models in the microbioreactor cell culture. The ratio between maximum and minimum cell number is 1.48 which shows a limited effect of culture condition on cell growth rate. The ratio of max to min for exosome content is 6.6 which is significantly higher than the cell proliferation ratio. The ratio of maximum to minimum for Glucose content is 2.8 which shows the constraint effect of the parameter in culture condition. Therefore, glucose concentration below 40 mg/dL is not allowed in the incubation period.

**Figure 3 cells-14-00676-f003:**
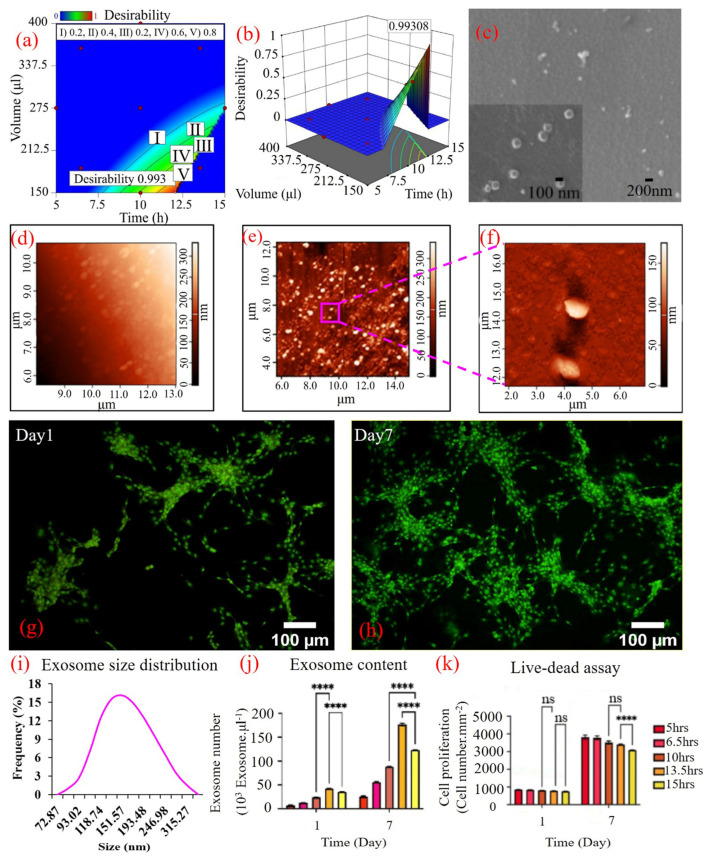
Microbioreactor culture optimization. Optimizing the cell culture condition in the micro-bioreactor is based on maximizing the exosome content while keeping the glucose content above the minimum level and the cells prolife in all time of cultivation. (**a**,**b**) 2D and 3D heat-map of Optimized point, respectively. Characterization of exosomes and cell proliferation in this figure. (**c**) Scanning electron microscopy (SEM) micrograph of the exosomes. The exosomes show a uniform size distribution of around 120–150 nm. (**d**) Atomic force microscopy (AFM) micrograph. Control slide shows no exosome on the glass slide. (**e**) AFM micrograph of exosome harvested from optimized culture condition in the microbioreactor. The exosomes are rounded-shaped structures showing dimensions of ranging 80 nm to 300 nm diameter. The image acquired in semi-contact scanning AFM with no enrichment or processing for exosome isolation. (**f**) Enlarged image showing two exosomes; (**g**,**h**) Fluorescent microscopy of live U87MG cells in µBR by AO staining on days 1 and 7 in 5 h exchange intervals (**i**) Size distribution analysis of exosome collected from µBR using dynamic light scattering (DLS). (**j**) Exosome content of culture medium in µBR at different hours of medium exchange intervals (5, 6.5, 10, 13.5 and 15 h) (**k**) Quantitative data of live-dead assay in µBR on days 1 and 7 at different hours of culture medium exchange intervals (5, 6.5, 10, 13.5 and 15 h). ns: no significant, and **** *p*-values < 0.0001.

**Figure 4 cells-14-00676-f004:**
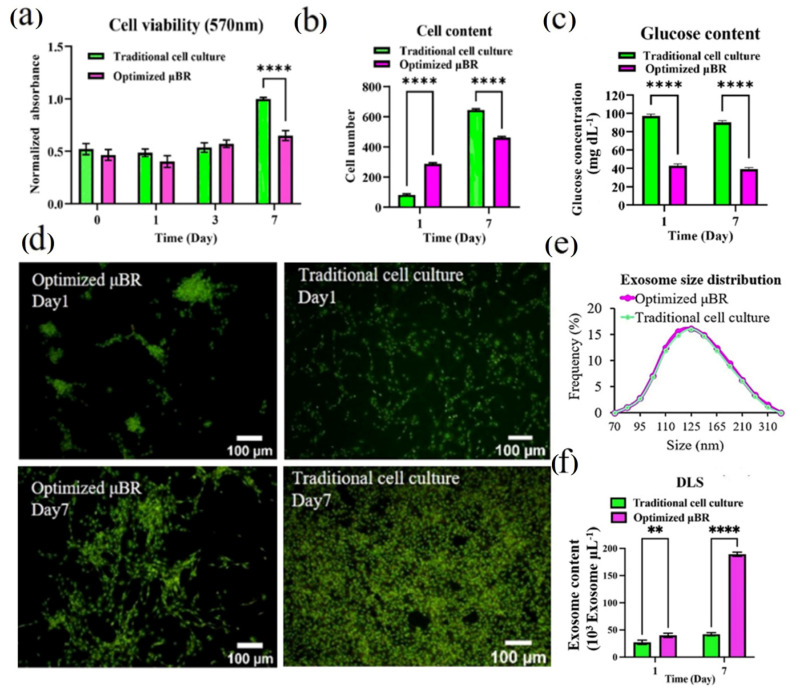
Glioblastoma cell culture under optimized condition in μBR vs. traditional cell culture. (**a**) Viability results of U87MG cell cultured in two groups (traditional cell culture and optimized µBR) for 7 days using MTT assay, (*p*-value < 0.01) shows the cells are prolife in the optimized condition in µBR. (**b**) The Quantified graph of Cell count in traditional cell culture and µBR on days 1 and 7. (**c**) The graph of Glucose content in traditional cell culture and optimized µBR on days 1 and 7. (**d**) Fluorescent microscopic image of live/dead assay on U87MG cells in traditional culture and optimized µBR by AO staining on days 1 and 7. (**e**) Size distribution analysis of exosome collected from Traditional cell culture and optimized µBR using dynamic light scattering (DLS). (**f**) Exosome content of optimized culture medium in µBR vs. Traditional cell culture on days 1 and 7, shows more than 4-fold accumulation of exosome in optimized µBR using optimized fed-batch strategy. ** *p*-values < 0.01, and **** *p*-values < 0.0001.

**Figure 5 cells-14-00676-f005:**
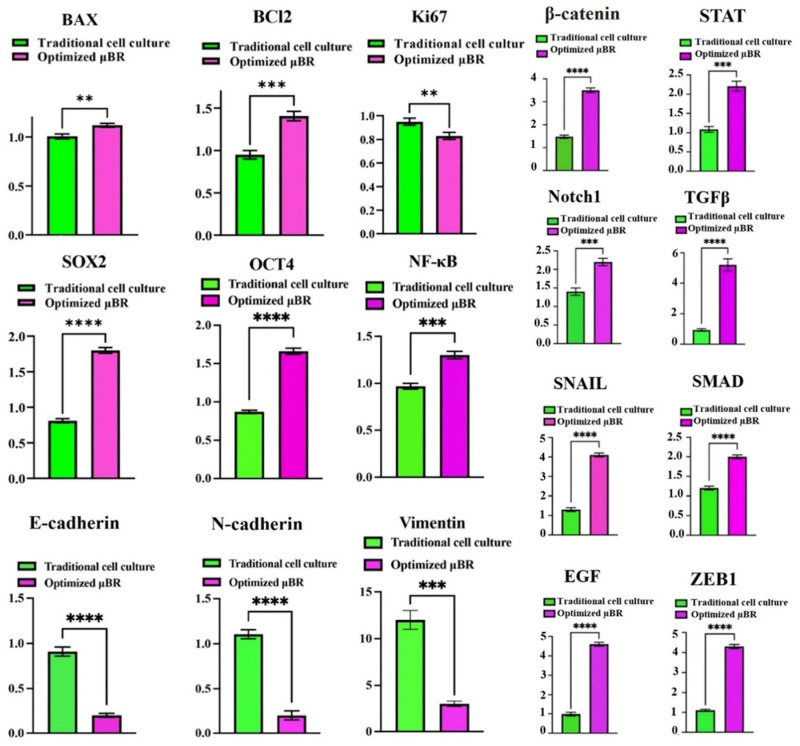
The RT-qPCR analysis. The mRNA expression levels of the *BAX* (as an apoptotic marker), *Bcl2* (as an anti-apoptotic marker), *Ki67* (as a proliferation marker), *OCT-4* and *SOX2* (as stemness gene), *NF-κB* (as an inflammation factor and coactivator of CSC and EMT related genes), *E-cadherin*, *N-cadherin*, *Vimentin*, *SNAIL* and *ZEB1* (as EMT marker), *β-catenin*, *STAT3*, *Notch1*, *TGFβ*, *EGF* and *SMAD* (as crucial regulator genes in stemness and EMT related genes) and *β-actin* (as an internal gene), were detected by Real-time PCR in µBR and traditional cell culture on days 0 and 7. ** *p*-values < 0.01, *** *p*-values < 0.001 and **** *p*-values < 0.0001.

**Figure 6 cells-14-00676-f006:**
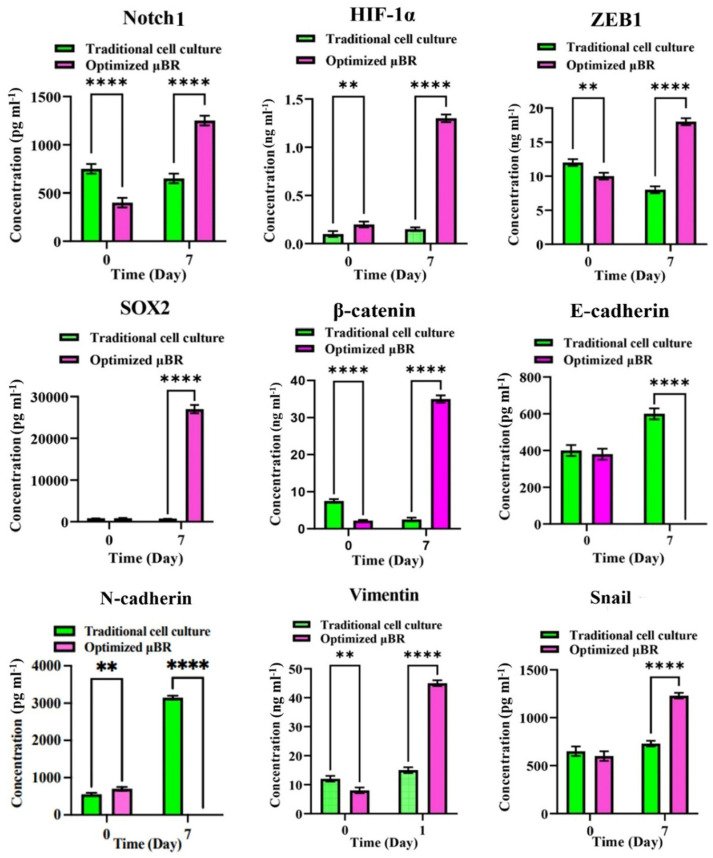
Enzyme-linked immunosorbent assay (ELISA). The expression level of key proteins in Glioblastoma cell line (U87) culture under optimized condition in µBR vs. Traditional cell culture supernatant were measured by ELISA. Notch1 (main player of important pathway in triggering EMT), HIF1 (Hypoxia-Inducible Factor 1 is known for its involvement in promoting aggressiveness in cancer cells, regulates cellular response to hypoxia and can enhance invasion potential by inducing various target genes related to angiogenesis, epithelial-mesenchymal transition (EMT), and extracellular matrix remodeling), ZEB1 (as EMT marker, related to dedifferentiation of the glioblastoma cells to mesenchyme which convert cells to a motile type and its expression is associated with higher grades of malignancy, tumor progression and invasion), SOX2 (as stemness marker has been linked to enhanced invasiveness in various cancer types, potentially through EMT induction and interaction with other signaling pathways), β-catenin (a component of the cell-cell adhesion complex and of the canonical Wnt pathway, regulates proliferation, adhesion, and migration in different cell types, as crucial regulator in stemness and EMT related genes), E-cadherin (an epithelial marker), N-cadherin (cancer epithelial marker), Vimentin (mesenchymal marker), Snail (as EMT marker constitutes a master switch that directly represses the epithelial phenotype). ** *p*-values < 0.01, and **** *p*-values < 0.0001.

**Figure 7 cells-14-00676-f007:**
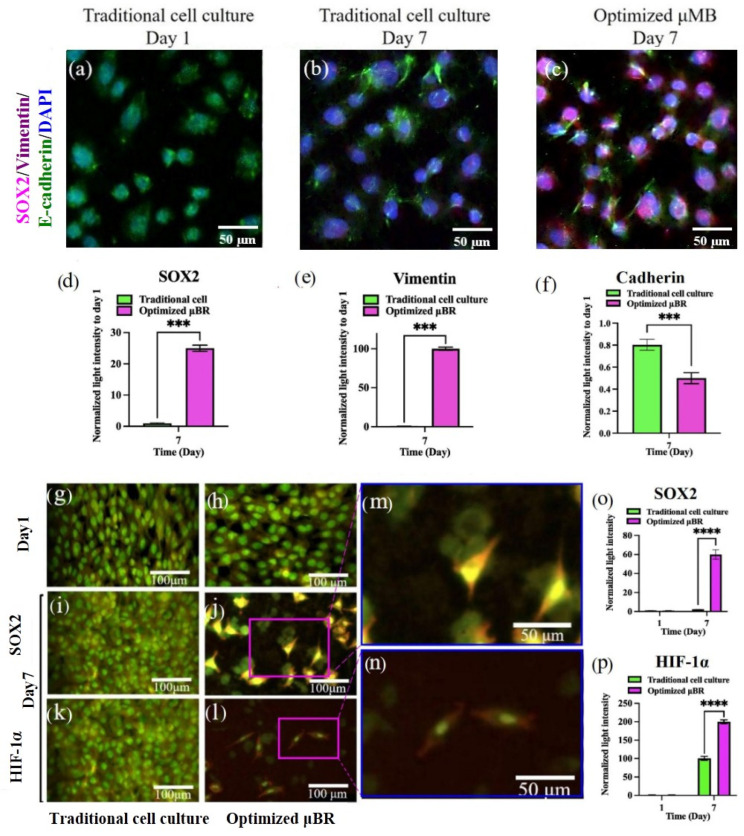
Comparative analysis of EMT and CSC marker proteins in Glioblastoma cell line culture under optimized condition in Microbioreactor vs. Traditional cell culture. (**a**–**c**) Immunofluorescence analysis for SOX2 (as a stemness marker) and Vimentin and E/N-Cadherin (as EMT marker) was performed in Glioblastoma cell line culture under optimized condition in µBR vs. Traditional cell culture on days 1 and 7. (**d**–**f**,**o**,**p**) Quantified graphs of immunofluorescence staining of EMT and CSC marker in Glioblastoma cell line (U87) in µBR vs. Traditional cell culture on days 1 and 7. (**g**–**n**) Fluorescent microscopic images of live U87MG cells in Traditional cell culture and optimized µBR by AO staining on days 1 and 7 ((**i**,**k**): same location with different filters). *** *p*-values < 0.001 and **** *p*-values < 0.0001.

**Figure 8 cells-14-00676-f008:**
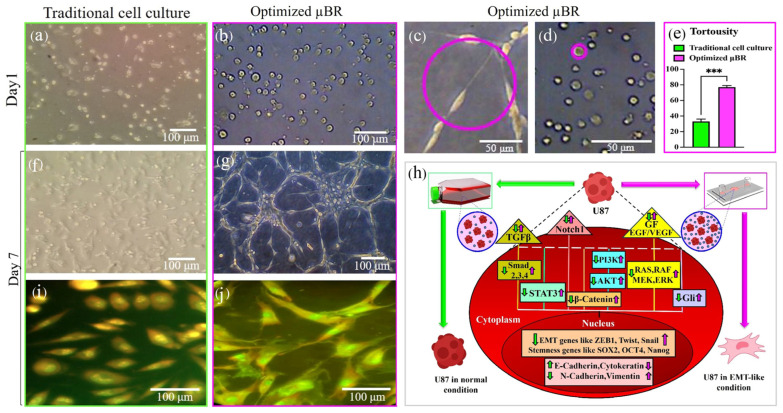
Mathematical modeling of Glioblastoma cell line culture in µBR and alteration of cell fate. (**a**–**j**) Behavior of Glioblastoma cell line (U87) in optimized µBR vs. Traditional cell culture. (**a**,**b**,**f**,**g**) Round and tortuous cells (U87) culture under optimized condition in µBR vs. Traditional cell culture on days 1 and 7. (**c**–**e**,**h**) Closer view of the round and tortuous cells in the optimized µBR on days 1 and 7. (**i**,**j**) Fluorescent microscopic image of the U87-MG cell line morphology in Traditional cell culture and optimized µBR by AO staining on day 7. (**e**) Quantitative analysis of the cell morphology changes (After identifying the outline of the cell, a peripheral circle was created to calculate the tortuosity index using Image J software). (**h**) Simplified scheme of possible critical proteins involved in signaling pathways in GSCs maintenance in µBR compared to traditional cell culture. All shades of green represent the U87 cell line under normal conditions in traditional cell culture, while all shades of purple indicate the U87 cell line in the optimized µBR inducing an EMT-like condition. *** *p*-values < 0.001.

**Table 1 cells-14-00676-t001:** Design matrix of medium optimization for cell proliferation exosome content, and glucose content.

Standard Order.	Variable		Results					
	A: Time (h)	B: Volume (µL)	Experimental Values			Predictive Values		
			Cell Proliferation (Cell Number/mm^2^)	Exosomes Content (10^3^ Exosomes/µL)	Glucose Content (mg/dL)	Cell Proliferation (Cell Number/mm^2^)	Exosomes Content (10^3^ Exosomes/µL)	Glucose Content (mg/dL)
1	6.46	186	3650	75	83.6	3420	74.88	75.79
2	13.54	186	2700	178	35.2	2941	177.58	36.68
3	6.46	363	3900	38	94.6	4089	37.92	99.05
4	13.54	363	3500	72	55	3610	71.62	59.93
5	5	275	3800	27	99	3854	27.03	95.52
6	15	275	3400	123	50.6	3176	123.47	40.21
7	10	150	2850	156	44	3042	156.28	51.42
8	10	400	4000	55	88	3988	55.22	84.31
9	10	275	3650	85	70.4	3515	81.40	67.86
10	10	275	3500	80	68.2	3515	81.40	67.86
11	10	275	3600	82	61.6	3515	81.40	67.86
12	10	275	3500	77	74.8	3515	81.40	67.86
13	10	275	3650	83	57.2	3515	81.40	67.86

**Table 2 cells-14-00676-t002:** Sequence of primers used in real-time RT-PCR.

Genes	Forward Primer Sequence (5′->3′)	Reverse Primer Sequence (5′->3′)
*BAX*	GGCCCTTTTGCTTCAGGGTT	GGAAAAAGACCTCTCGGGGG
*BCL2*	GGTGAACTGGGGGAGGATTG	ATCACCAAGTGCACCTACCC
*Ki-67*	TTTGGGTGCGACTTGACGAG	CGTCCAGCATGTTCTGAGGA
*OCT-4*	CGCCGTATGAGTTCTGTGGG	CTGATCTGCTGCAGTGTGGGT
*SOX2*	ATGGACAGTTACGCGCACAT	CGAGCTGGTCATGGAGTTGT
*NF-* *κ* *B*	CGACAGCGGGGAAAGACAC	TGCCATTCTGAAGCTGGTGG
*E-cadherin*	GCTGGACCGAGAGAGTTTCC	CAAAATCCAAGCCCGTGGTG
*N-cadherin*	AAAGACCCATCCACGCTGAG	GCTCAAGGACCCCAAGGTG
*Vimentin*	GGACCAGCTAACCAACGACA	AAGGTCAAGACGTGCCAGAG
*SNAIL*	CGAGTGGTTCTTCTGCGCTA	GGGCTGCTGGAAGGTAAACT
*ZEB1*	GGCGCAATAACGGAAAGGAAG	AGCCAGAATGGGAAAAGCGT
*β-catenin*	GGAGGAAGGTCTGAGGAGCA	AGGCTCCAGAAGCAGTCATC
*STAT*	CAGGAGCTGAAAAACCAGCAGT	GGGGATTCGGGGATAGAGGA
*Notch1*	GCGAGGAAGATACGGAGTGG	GCCTTCCAGCCTGCCTTTTA
*TGFβ*	TGGTGGAAACCCACAACGAA	CGGTAGTGAACCCGTTGATG
*EGF*	CTGAATGTCCCCTGTCCCAC	TGCATTGACCCCAAGGTTGA
*SMAD*	TCACATCTCTCCCGTGCTGC	CATGCAGTGAGGCAATCGAC
*β-actin*	GGCATCCTCACCCTGAAGTA	AGGTGTGGTGCCAGATTTTC

## Data Availability

All relevant data are included in the article and its Appendix A.

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
