# Peer review of "Optimization of In-Situ Exosome Enrichment Methodology On-a-Chip to Mimic Tumor Microenvironment Induces Cancer Stemness in Glioblastoma Tumor Model"

_cells, 2025, doi:10.3390/cells14090676_

Round 1
Reviewer 1 Report
Comments and Suggestions for Authors
The authors have taken a logical approach to an existing problem and have delivered well the outcomes and conclusions. It was an interesting read where the authors develop a microfluidic Bioreactor to mimic tumor microenvironment with quite a success. The microBR could produce hypoxic environment with reduced proliferation and increased EMT which is the characteristic of solid tumors. Future developments on the same kind of reactors where EMT and cancer stemness parameters can be further improved and interactions with other cells like immune cells are shown is altogether worth attempting.
I have the following minor comments:
1) The only part which was not clear to me as a reviewer are the lines 88-94. The authors should explain it better with citations. There are no references to these lines.
2) There is a typographic error in line 188 where it should be 'siRNAs'
Author Response
Comments and Suggestions for Authors
The authors have taken a logical approach to an existing problem and have delivered well the outcomes and conclusions. It was an interesting read where the authors develop a microfluidic Bioreactor to mimic tumor microenvironment with quite a success. The microBR could produce hypoxic environment with reduced proliferation and increased EMT which is the characteristic of solid tumors. Future developments on the same kind of reactors where EMT and cancer stemness parameters can be further improved and interactions with other cells like immune cells are shown is altogether worth attempting.
I have the following minor comments:
Comments 1) The only part which was not clear to me as a reviewer are the lines 88-94. The authors should explain it better with citations. There are no references to these lines.
Response 1: We appreciate the reviewer’s insightful comment regarding lines 88–94. In response, we have revised the text to improve clarity and added appropriate references [35,36], to support the statements made in this section. Furthermore, we would like to clarify that the content of this part is based on the experimental design considerations, the detailed calculations and methodology of which have been included in the Supplementary Information. These additions aim to provide a comprehensive explanation for the reviewer and readers. Thank you again for your valuable feedback.
Previous version: Understanding that the thermodynamic activity of almost all cell products is strongly influenced by their effective concentrations in the TME, we performed basic measurements on the spatial conformation of cancer cells in the TME. The distance between cancer cells in a GBM solid tumor is less than 2 µm, with cell diameters ranging between 15 to 20 µm and a height of 10 µm. This means most of the volume of the solid tumor is occupied by cells. Thus, at best, about 29% of the volume of a solid tumor is interstitial space, which acts as a TME reservoir extracellular matrix.
Revised version: Understanding that the thermodynamic activity of almost all cell products is strongly influenced by their effective concentrations in the TME [35,36], we performed basic measurements on the spatial conformation of cancer cells in the TME. The distance between cancer cells in a GBM solid tumor is less than 2 µm, with cell diameters ranging between 15 to 20 µm and a height of 10 µm. This means most of the volume of the solid tumor is occupied by cells. Thus, at best, about 29% of the volume of a solid tumor is interstitial space, which acts as a TME reservoir extracellular matrix (See supplementary material).
- Lyssiotis, C.A.; Kimmelman, A.C. Metabolic Interactions in the Tumor Microenvironment. Trends in Cell Biology 2017, 27, 863–875, doi:10.1016/j.tcb.2017.06.003.
- Heiden, M.G. Vander Understanding the Warburg Effect . Science 2009, 324, 1029-1033, doi:10.1126/science.1160809.
Comments 2) There is a typographic error in line 188 where it should be 'siRNAs'
Response 2: We thank the reviewer for pointing out the typographic error in line 188. The term has been corrected to “siRNAs” in the revised manuscript, and the correction has been highlighted in the text for clarity.
Previous version: Each exosome carries several miRs, iRNAs, functional proteins and peptides envel-oped in a bilayer membrane and secreted out of cell via exocytosis [34].
Revised version: Each exosome carries several miRs, siRNAs, functional proteins and peptides enveloped in a bilayer membrane and secreted out of cell via exocytosis [40].
Reviewer 2 Report
Comments and Suggestions for Authors
We would like to thank the authors for a well-designed and well-written paper. Keep up the good work.
Title: Should the word device or method " be added? You wrote Optimization of in-situ exosome enrichment on-a-chip...
It should read (Optimization of in-situ exosome enrichment test on-a-chip) or
(Optimization of in-situ exosome enrichment method on-a-chip) or
(Optimization of in-situ exosome enrichment device on-a-chip)n, or something to that effect.
Abstract:
All good
Intro
All good with the proper references.
Materials and methods
Lines 87 and 88 needed to be merged together.
Line 243, where the sole equation is presented (When stating mathematical equations, usually they are numbered 1,2,3) This equation should end with the numenbr (1).
The figures presented are all good.
The tables given are good.
Conclusion:
Why are there no conclusions for the large amount of data you gathered?
References:
What type of reference is ref (37)? There was no journal name, authors, etc. Do you really need to reference this method?
42. Cells, O. Structural and Mechanical Characteristics of Exosomes from o u r n a l a Me Structural and Mechanical Characteristics of Exosomes. 2021, doi:00.0000/00000000.J.
(What type of reference is that?) Please rewrite and fix this.
Please go over all references one by one and make sure the volume number, page numbers, and author's name are correct.
best,
Author Response
Comments and Suggestions for Authors
We would like to thank the authors for a well-designed and well-written paper. Keep up the good work.
Comment 1) Title: Should the word device or method " be added? You wrote Optimization of in-situ exosome enrichment on-a-chip...
It should read (Optimization of in-situ exosome enrichment test on-a-chip) or
(Optimization of in-situ exosome enrichment method on-a-chip) or
(Optimization of in-situ exosome enrichment device on-a-chip), or something to that effect.
Response 1: We appreciate the reviewer’s valuable suggestion regarding the clarity of the manuscript title.
Previous version: Optimization of in-situ exosome enrichment on-a-chip to mimic tumor microenvironment induces cancer stemness in glioblastoma tumor model
As recommended, we have revised the title to:
Revised version: "Optimization of in-situ exosome enrichment methodology on-a-chip to mimic tumor microenvironment induces cancer stemness in glioblastoma tumor model."
We believe this revised title better reflects the content and scope of our study.
Abstract: All good
Intro: All good with the proper references.
Materials and methods
Comment 2) Lines 87 and 88 needed to be merged together.
Response 2: We thank the reviewer for the observation. However, line 88 is a subsection title ("2.1.1. Estimating the Local Concentration of a Chemical Secreted from Tumor Cells within the TME"), and as such, it is needed to remain structurally separate from the following paragraph according to the journal’s formatting guidelines.
Comment 3) Line 243, where the sole equation is presented (When stating mathematical equations, usually they are numbered 1,2,3) This equation should end with the number (1).
Response 3: We appreciate the reviewer’s suggestion. The equation has now been revised to include the appropriate numbering and ends with the label (1) as recommended.
Previous version: The coefficients in the first and second-order polynomials were calculated using multiple regression analysis of the obtained results experimentally.
Revised version: The coefficients in the first and second-order polynomials were calculated using multiple regression analysis of the obtained results experimentally (See equation (1)).
The figures presented are all good.
The tables given are good.
Comment 4) Conclusion:
Why are there no conclusions for the large amount of data you gathered?
Response 4: Thank you for the subtle notice of missing part in subtitles. Actually, we have written the very last two paragraphs of the article as our conclusion of this article, covering key features of the study followed by significance of observations and future directions for further studies and possible collaborations. Based on that the subtitle of conclusion is added to the manuscript preventing redundancy.
- Conclusion
Microfluidic device application in in-vitro diseased models has been drawing researchers’ attention in the past two decades to address their concerns about mimicking pathophysiological conditions. Specifically, in the realm of in-vitro tumor models, TME is the focal point of reasoning to apply microfluidic systems. This research has shown that although downsizing of the culture environment by microsystems is a leap towards TME simulation, the effective chemical activity of components in the interstitial space of tumor cells is significantly higher in native TME. Applying RSM in this research, we approached the highest possible accumulation of components in the TME, which is still lower than native TME in patients’ bodies.
The results enlighten that through a fed-batch strategy of medium renewal in µBR with a vast surface, the microsystem can revive the plasticity of the glioblastoma cell line with no additional conditioning components. To shed light on how far this plasticity can go, future investigations are needed, especially in some cases where long appendages in cells (axon-like) have been observed, suspicious of differentiation of CSC-like cells towards neurons. Another future direction can include single-cell RNA sequencing to elaborate dominant pathways beside native tumor biopsy culture, which can detect trace effective components that normally diluted out in the culture medium. The coculture of this model with immune cells, either brain microglial cells or chemokines secreted from blood-circulating immune cells, can also be highly interesting for a wide variety of research investigators.
Comment 5) References:
What type of reference is ref (37)? There was no journal name, authors, etc. Do you really need to reference this method?
- Cells, O. Structural and Mechanical Characteristics of Exosomes from o u r n a l a Me Structural and Mechanical Characteristics of Exosomes. 2021, doi:00.0000/00000000.J.
(What type of reference is that?) Please rewrite and fix this.
Please go over all references one by one and make sure the volume number, page numbers, and author's name are correct.
Response 5: Thank you for pointing that out. Reference (37) has been removed from the manuscript and we referred to it as a catalog number in the Methods section based on the manufacturer's instructions.
Revised version: The glucose levels in two groups, optimized micro-bioreactor (optimized µBR) and the Traditional cell culture, were detected using glucose assay kit (Ab65333).
Additionally, all remaining references have been thoroughly reviewed and edited to ensure they include accurate author names, journal titles, volume and issue numbers, page ranges, and DOIs where applicable.
Revised version: Yurtsever, A.; Yoshida, T.; Behjat, A.B.; Araki, Y.; Hanayama, R.; Fukuma, T. Structural and Mechanical Characteristics of Exosomes from Osteosarcoma Cells Explored. 2021, 6661–6677, doi:10.1039/d0nr09178b.

Reviewer 3 Report
Comments and Suggestions for Authors
The authors are encouraged to expand the introduction section by including additional key studies relevant to the use of microfluidic devices for Extracellular Vesicles research. For example, the following works could provide contextual insights to this manuscript:
1. Zhu Q, Heon M, Zhao Z, He M. Microfluidic engineering of exosomes: editing cellular messages for precision therapeutics. Lab Chip. 2018;18(12):1690-1703. doi:10.1039/c8lc00246k
2. Jo W, Jeong D, Kim J, et al. Microfluidic fabrication of cell-derived nanovesicles as endogenous RNA carriers. Lab Chip. 2014;14(7):1261-1269. doi:10.1039/c3lc50993a
3. Wu Y, Wang Y, Lu Y, et al. Microfluidic Technology for the Isolation and Analysis of Exosomes. Micromachines (Basel). 2022;13(10):1571. Published 2022 Sep 22. doi:10.3390/mi13101571
4. Kwong Hong Tsang D, Lieberthal TJ, Watts C, et al. Chemically Functionalised Graphene FET Biosensor for the Label-free Sensing of Exosomes. Sci Rep. 2019;9(1):13946. Published 2019 Sep 26. doi:10.1038/s41598-019-50412-9
5. Guo SC, Tao SC, Dawn H. Microfluidics-based on-a-chip systems for isolating and analysing extracellular vesicles. J Extracell Vesicles. 2018;7(1):1508271. Published 2018 Aug 20. doi:10.1080/20013078.2018.1508271
6. Sariano PA, Mizenko RR, Shirure VS, et al. Convection and extracellular matrix binding control interstitial transport of extracellular vesicles. J Extracell Vesicles. 2023;12(4):e12323. doi:10.1002/jev2.12323
7. Nguyen VVT, Ye S, Gkouzioti V, et al. A human kidney and liver organoid-based multi-organ-on-a-chip model to study the therapeutic effects and biodistribution of mesenchymal stromal cell-derived extracellular vesicles. J Extracell Vesicles. 2022;11(11):e12280. doi:10.1002/jev2.12280
Their inclusion would help to demonstrate the study’s novelty.
Author Response
Comment 1) The authors are encouraged to expand the introduction section by including additional key studies relevant to the use of microfluidic devices for Extracellular Vesicles research. For example, the following works could provide contextual insights to this manuscript:
- Zhu Q, Heon M, Zhao Z, He M. Microfluidic engineering of exosomes: editing cellular messages for precision therapeutics. Lab Chip. 2018;18(12):1690-1703. doi:10.1039/c8lc00246k
- Jo W, Jeong D, Kim J, et al. Microfluidic fabrication of cell-derived nanovesicles as endogenous RNA carriers. Lab Chip. 2014;14(7):1261-1269. doi:10.1039/c3lc50993a
- Wu Y, Wang Y, Lu Y, et al. Microfluidic Technology for the Isolation and Analysis of Exosomes. Micromachines (Basel). 2022;13(10):1571. Published 2022 Sep 22. doi:10.3390/mi13101571
- Kwong Hong Tsang D, Lieberthal TJ, Watts C, et al. Chemically Functionalised Graphene FET Biosensor for the Label-free Sensing of Exosomes. Sci Rep. 2019;9(1):13946. Published 2019 Sep 26. doi:10.1038/s41598-019-50412-9
- Guo SC, Tao SC, Dawn H. Microfluidics-based on-a-chip systems for isolating and analysing extracellular vesicles. J Extracell Vesicles. 2018;7(1):1508271. Published 2018 Aug 20. doi:10.1080/20013078.2018.1508271
- Sariano PA, Mizenko RR, Shirure VS, et al. Convection and extracellular matrix binding control interstitial transport of extracellular vesicles. J Extracell Vesicles. 2023;12(4):e12323. doi:10.1002/jev2.12323
- Nguyen VVT, Ye S, Gkouzioti V, et al. A human kidney and liver organoid-based multi-organ-on-a-chip model to study the therapeutic effects and biodistribution of mesenchymal stromal cell-derived extracellular vesicles. J Extracell Vesicles. 2022;11(11):e12280. doi:10.1002/jev2.12280
Their inclusion would help to demonstrate the study’s novelty.
Response1: Thank you for your valuable suggestion. The introduction section has been expanded to include additional key studies relevant to the application of microfluidic devices in extracellular vesicle (EV) research. These references have been integrated to provide a broader context and support the rationale for the present study.
Introduction
The lack of a comprehensive in-vitro tumor model is perhaps the deepest deficit in the realm of cancer study and therapeutic development. Animal models [1,2], even patient-derived xenograft (PDX) models [3,4], suffer from humane considerations and lack accuracy due to genotypical and, more importantly, immunological differences between species. Additionally, they offer very low throughput feasible assays.
Recent elegant studies on in-vitro 3D tumoroid models derived from patient biopsies [5,6] in microfluidic systems [7,8] to simulate the tumor microenvironment (TME) [9,10] using geometric varieties in the bioreactor with different mass transfer strategies [11,12], such as air-liquid interface feeding systems[13,14], have shown promise. However, there is still no reliable in-vitro tumor model available to re-searchers in this field. It is evident that a system intended to integrate all these ad-vancements requires sophisticated instrumentation and specialties. Nevertheless, tumor microenvironment mimicry remains the deepest common root of all these efforts.
Access to an in-vitro tumor model that encompasses all features of a tumor within the body is limited by several constraints. Among them, the chemically complex and heavily dense tumor structure is the most significant [15,16]. It is be-lieved that cancer cells, through the TME—a chemically complex medium [17] in-undated with chemokines [18] and extracellular microvesicles [19]—communicate with other cells both in their vicinity and distally to create an environment favorable for cancer progression [20].
The complexity of the TME is so extreme that it often sends mixed signals. For example, inducing angiogenesis [21], a cancer hallmark [22,23], results in incomplete tumor microvasculature formation, which is incapable of sufficient oxygenation of the tumor cells. Consequently, many tumor cells suffer from hypoxia [24] to such an extent that the hypoxic condition becomes a preconditioning for inducing tumor aggressiveness [25]. In many research projects, conditioned medium has been used to mimic the TME [26,27], and in others, TME simulation is targeted using microfluidic [28]or 3D tumoroids [29,30]. However, tumor models still cannot behave the same way they do in patient's bodies.
We hypothesized that in either microfluidic systems or traditional tumor cell culture in conditioned medium, the TME that cancer cells try to provide for them-selves has a far lower concentration of chemokines secreted by the cells compared to the native cancer cells within the TME in vivo. To test this hypothesis, glioblastoma multiforme (GBM) cancer was modeled in a micro-bioreactor (μBR) in 2D culture without conditioned medium. The elimination of all systemic complexities, such as 3D organoid culture, intricate geometry, or engineered feeding systems like air-liquid interface, not only makes the tumor model more accessible in tumor research labs but also clarifies that controlled-TME condition is the essence of in-vitro tumor modeling.
To introduce a quantifiable parameter as a criterion for microenvironment enrichment in the μBR model, the exosome content of the culture medium was selected. This choice was based on a growing body of research highlighting the role of extracellular vesicles—particularly exosomes—as key mediators in cancer progression, due to their ability to carry a complex cargo of diverse chemicals, RNAs, and functional proteins that facilitate intertumoral communication [31–34].
To address all aforementioned factors in one simple model, GBM cells were cultured in a vast surface area in a microfluidic bioreactor, where the TME cultivation conditions were optimized via response surface methodology (RSM) and compared with traditional cell culture.
- Sariano, P.A.; Mizenko, R.R.; Shirure, V.S.; Brandt, A.K.; Nguyen, B.B.; Nesiri, C.; Shergill, B.S.; Brostoff, T.; Rocke, D.M.; Borowsky, A.D.; et al. Convection and Extracellular Matrix Binding Control Interstitial Transport of Extracellular Vesicles. Journal of Extracellular Vesicles 2023, 12, doi:10.1002/jev2.12323.
- Nguyen, V.V.T.; Ye, S.; Gkouzioti, V.; van Wolferen, M.E.; Yengej, F.Y.; Melkert, D.; Siti, S.; de Jong, B.; Besseling, P.J.; Spee, B.; et al. A Human Kidney and Liver Organoid-Based Multi-Organ-on-a-Chip Model to Study the Therapeutic Effects and Biodistribution of Mesenchymal Stromal Cell-Derived Extracellular Vesicles. Journal of Extracellular Vesicles 2022, 11, doi:10.1002/jev2.12280.
- Jo, W.; Jeong, D.; Kim, J.; Cho, S.; Jang, S.C.; Han, C.; Kang, J.Y.; Gho, Y.S.; Park, J. Microfluidic Fabrication of Cell-Derived Nanovesicles as Endogenous RNA Carriers. Lab on a Chip 2014, 14, 1261–1269, doi:10.1039/c3lc50993a.
- Zhu, Q.; Heon, M.; Zhao, Z.; He, M. Microfluidic Engineering of Exosomes: Editing Cellular Messages for Precision Therapeutics. Lab on a Chip 2018, 18, 1690–1703, doi:10.1039/c8lc00246k.
